# Identifying Socioeconomic Determinants of Households' Forest Dependence in the Rubi-Tele Hunting Domain, DR Congo: A Logistic Regression Analysis

**Richard K. Mendako** [1], **Gang Tian** [1,*] **and Patrick M. Matata** [2]

1   College of Economics & Management, Northeast Forestry University, Harbin 150040, China
2   Faculty of Economics and Management, University of Kisangani, Kisangani 2012, Democratic Republic of the Congo
*   Correspondence: tiangang_nefu@126.com; Tel.: +86-1894-6059-819

**Abstract:** Rural households depend on forest resources for cash and subsistence needs. Thus, forests represent a valuable natural capital for the rural economy, particularly in developing countries. However, depending on various factors, there are dissimilarities in the rural livelihoods' reliance on forests. Therefore, this study attempted to determine and characterize the level of forest dependence and identify the demographic and socioeconomic factors influencing the households' dependence on forests in the Rubi-Tele Hunting Domain (RTHD)/Democratic Republic of the Congo (DR Congo). Demographic and socioeconomic data of forest dependents and other qualitative information were collected through structured household-level surveys, focus group discussions, and key informant interviews. Descriptive statistical analysis, Kruskal–Wallis Test, $\chi^2$ test of independence, and binary logistic regression model were used for data analysis. The findings revealed that the forest dependence index varied from 0.01 to 1 (Mean = 0.46, SD = 0.30). The distribution of forest-dependent households by wealth status (income tercile) and level of forest dependence differs significantly. Logistic regression revealed that household size and non-forest income were significant determinants of forest dependence and had the theoretically expected signs. The household size was in a positive association with forest dependence. Large families tended to depend more on forest resources. On the other hand, the non-forest income was inversely related to forest dependence, implying that forest-dependent households with non-forest income sources were less dependent on forest resources extraction. Other selected variables were not statistically significant while showing positive and negative associations with the reliance on forests. Achieving the balance between forest dependence and conservation requires promoting incentive policies to diversify livelihood opportunities and environmental education.

**Keywords:** forest use; forest income; forest dependence; rural livelihoods; logistic regression model; conservation; Rubi-Tele Hunting Domain; DR Congo

## 1. Introduction

It is widely acknowledged that biological diversity plays a key role in maintaining ecosystem sustainability, promoting human welfare, and enhancing economic development [1,2]. Since immemorial time, forests and resources they provide have been crucial to maintaining livelihoods [3–5], especially for populations that depend on them and are in extreme poverty [6,7]. The nation's economy and people's livelihoods both rely heavily on forest resources [8]. Local populations rely on forest resources for a variety of products, including fuelwood, construction materials, food, and medicine [9,10]. Over 2 billion people worldwide rely on forest resources for their livelihood [11]. By reducing inequality and supplying vital nutrients, food, and income, the extraction of forest resources, particularly non-timber forest products (NTFPs), is still essential for poor households in the majority

of developing countries [5,6,12–20]. This, despite rapid advancements in agriculture and the economy [17,21].

It is not just the world's poor rely on forests. Forests provide various benefits to rural residents throughout the developing world [22–25]. In addition to acting as safety nets and insurance against risks and shocks [26–30]; forests support rural livelihoods by bridging the seasonal income gap during times of low agricultural output and seasonal deficits [22,23,29,31,32]; eradicating poverty [33,34]; and promoting economic wellbeing [14,35,36]. Numerous research on the relationship between forests and livelihoods have shown how important forests are for sustaining and diversifying livelihoods and reducing poverty [3,5,37,38]. Other studies underlined that their ability to contribute to reducing poverty might be smaller than their ability to prevent and alleviate poverty [20,28]. Designing efficient forest management to minimize adverse effects on resources and guarantee people's wellbeing still depends on understanding the role of forest products in livelihoods and the factors that affect peoples' needs [13,19,20,28,31,39–46].

According to a study that examined 51 case studies from 17 developing nations, the income from the forest made up around 22% of all household income [37]. In South America, the percentage of forest revenue to overall household income varied between 14 and 20% [38,47]. In Asia, the contribution of forest income to total household income ranged from 10% to 20% [5]. This contribution varied between 30 and 45% of the total household income in other places [15,48]. Forest products make up over 35% of the total household income in Zimbabwe's rural areas [49]. According to large-scale income research conducted in the Congo basin by the Centre for International Forestry Research (CIFOR), households that live in or near forests often get between one-fifth and one-fourth of their income from forest-based activities [50,51]. In rural area of Malawi, forest resources represented 30% of total household income [14]. According to research carried out in the Ethiopian district of Dendi, 39% of household income was derived from forest resources [3]. In Nepal's lower Mustang forest, 22% of total annual household income was derived from forest-based activities [52]. Over 50% of total household income in Jharkhand, India, came from forest products [17,53].

The research mentioned above examined how forests affect the livelihoods of those living in and around them. The local population's reliance on forest resources should be considered while developing management plans and restoration procedures that affect forests. Identifying the factors influencing forest reliance is also crucial [40,41,54–56]. Households do not benefit at the same level from natural patrimony [51]. Forest reliance varies geographically among various socioeconomic categories [4,39,41,56–60]. Researchers and academicians are very interested in understanding the factors that influence household reliance on forest resources for the sustainability of forest resource management and biodiversity conservation, as evidenced by the relationship between forest dependency and biodiversity conservation [21,40,41,61]. The difference in households' reliance on the forest is essential [47]. The dependency of rural livelihood on forest variations arises because of dissimilarities in demographic factors, socioeconomic conditions, beliefs, and local community standards [2,4,17,47,54,55,58,62–70].

In some cases, other income sources help reduce dependence [41]. A study conducted in Cameroon showed that hunting requires technical know-how and stamina, which are more readily available to young men. In contrast, artisanal logging requires greater wealth and social relationships. Gathering typically requires a lot of labor and is, therefore more accessible to larger households [44,71,72]. The household size, ethnicity, household head's age and sex have been shown in some research to be significant [7,14,73,74].

Contrary to higher education, other research found a positive relationship between household size and forest dependency. Larger families rely more on forest resources because they have greater subsistence demands [3,41]. Similar research revealed a positive association between household heads' age and reliance on forest resources, despite the impact declining once the peak physical strength is reached [75]. Older people may become increasingly dependent on forests since they may have strong environmental knowledge

of their surroundings [21]. Logistic regression was used in another study to identify and analyze the potential causes of households' reliance on forests [44]. In Botswana's Kasane Forest Reserve, there was a positive and substantial correlation between forest dependence and family size. Households with more assets, however, were less reliant on forest resources [2]. Other studies discovered that the number of livestock and the size of farmland per household were important predictors of dependency [55]. Access to villages and agricultural revenue was the primary explanations for Cameroon's variations in forest income [51]. According to other studies [56,76], having a higher education is linked to using less forest resources. This is relevant because education provides additional chances for alternate livelihoods that could offer greater profits than activities involving the exploitation of forests [39]. An increase in family head education, non-forest income, and landholding were all found to reduce reliance on the forest [21]. A comparable study reported that household dependency on forest resources was significantly influenced by age, education, household size, livestock income, agriculture income, and off-farm income [17].Designing conservation and development approaches require understanding local communities and resource use patterns [2,41,77,78]. In the Congo Basin, national parks and other forest reserve landscapes are inhabited by people who rely on those ecosystems for multiple uses [19,55,71,79–81]. People-park conflicts may arise due to enforcing laws without recognizing the complex relationship between inhabitants and the environment in which they live [39,55,66]. Rubi-Tele Hunting Domain (RTHD) is a protected area in the northeast of the Democratic Republic of the Congo (DR Congo) where rural dwellers live. It is impossible to ignore the vital role that forest resources play in meeting the needs of those people, particularly through the informal sector and strategies for alleviating poverty [82]. However, there is a large knowledge gap in the RTHD on the resource utilization pattern. In the DR Congo, the economy based on gathering forest products, on which more than 70% of the population depends, is never seen as a natural remedy to poverty.

This study makes an important contribution to the research literature. The main goal is to investigate the factors influencing households' reliance on forests in the RTHD, DR Congo. Thus, the study (1) determines and characterizes the level of rural households' reliance on forest resources in the RTHD/DR Congo, and (2) identifies and analyzes the major socioeconomic and demographic variables influencing households' reliance on forest in this protected area. Following the new conservation paradigm based on local development, the study's findings are essential for establishing pro-poor development policies for conservation in the RTHD [28,31,79,83–93].

## 2. Materials and Methods

### 2.1. Study Area and Sampling

The study was conducted in RTHD, located in the Bas-Uélé Province. This province lies in the northern region of DR Congo, on the borders with South Sudan and the Central African Republic, and has a total area of 148,331.00 km$^2$. Order N°51/Agri./12 December 1930 established the RTHD between the Rubi and Tele rivers and Order N°64/Agri./ 28 November 1932 modified [94]. This protected area, which has an extent of more than 8000.00 km$^2$, was established as a "Hunting reserve" only six years after the establishment of Virunga National Park. The RTHD comprises 9080.00 km$^2$ of land, according to the World Database on Protected Areas (WCMC, UNEP). A hunting ban is the main management measure applied in this area (Article 2 of the Legal Act of 1930). Additionally, Articles 2 and 3 of the Legal Act of 1930 contain additional provisions that allowed the lifting of the hunting restriction for specific species subject to authorization or permits provided to various groups of beneficiaries, including rural households. The preservation of huge fauna is the main management goal of this protected area [95].

The Rubi-Tele Forest is characterized by *Gilbertiodendron Dewevrei*. It is a tropical primary forest species with attributes that prevent it from regenerating naturally. RTHD, generally, has small populations of fauna. In this Domain, a variety of large faunal species are particularly vulnerable to hunting and environmental deterioration. Chimpanzees

and small ungulates are also found in this ecosystem. Additionally, RTHD is home to species that serve as symbols, such the Okapi and the African Elephant. The presence of three lakes inside this protected area illustrates its function in conserving particular aquatic animals. Huge difficulties are currently being faced in this protected area. The local populations, hunters, and diamond dealers—many of whom frequently travel from nearby territories—pose threats from all sides.

The Territories of Aketi, Bambesa, and Buta, located in the Bas-Uélé Province, and the Territories of Banalia and Basoko, which are found in the Tshopo Province, are among the five Administrative Territorial Entities that the RTHD covers. It is dispersed over eight sectors in the Bas-Uélé Province, including Bayeu-Bogbama, Mabinza, Makere II, Mongazulu, Yoko, and others in the Tshopo Province, including Baboa of Kole, Wahanga, and Yamandundu. The ICCN (Congolese Institute for Nature Conservation)/Provincial Office conducted research on the delimitation of the RTHD in the past and discovered that 3731 people had occupied the area, dispersed among 241 camps and 23 villages. A total of 674 individuals lived in the 241 camps (632 inside the domain and 42 outside), and it was estimated that 3057 more people lived in 23 settlements (12 outside and 14 inside) [95]. One of the residents living in the RTHD and the periphery's main activities is hunting for subsistence use and sale. These populations also engage in agriculture, fishing, and resource gathering from the forest. According to the study above, the RTHD, mining squares, and forest titles are superimposed.

Eight easily accessible villages in the RBHD were randomly chosen for sampling. Red small stars mark these surveyed villages in Figure 1 (Baangba, Bongbongo, Bobusanga, Ngbete, Bondeme, Boyo, Bombole, and Benge). Simple random sampling was used to choose the households. A total of 127 households were then sampled from the specified villages. A total of 114 among them had positive forest income. Data from these households were thus used in this study's analysis.

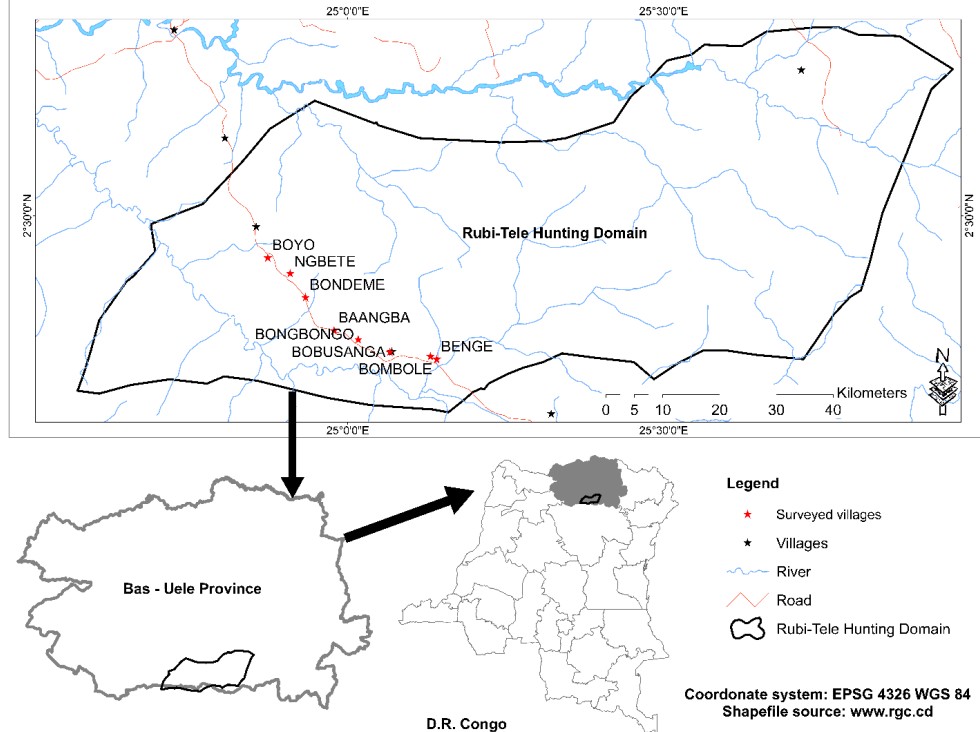

**Figure 1.** The study area's map.

## 2.2. Questionnaire Design and Data Collection

Primary data were collected using structured household-level surveys, focus groups, and key informant interviews. We developed and modified the questionnaire following the

Poverty–Environment Network (PEN) [96]. The questionnaire collected basic household demographic, socioeconomic data, and other qualitative information. Quantitative information on household cash and non-cash income was gathered from all main livelihood strategies. There were both closed and open questions in the structured household survey. The questionnaire was first tested among a few households in the rural part of Kisangani City. Finally, from November to December 2020, we administered it to 127 households while conducting fieldwork in the RTHD. The final sampling unit was the household, defined as "a group of individuals living under the same roof and pooling resources for their livelihood" [96]. Surveys were conducted with household heads. In the absence of the latter, the questionnaire was administered to senior household members. Both were occasionally interviewed to ensure the most accurate recall of the answers [69]. Two qualified researchers assisted the surveys.

The study area status and common household data were collected through focus groups and key informant interviews. An extensive review of published and unpublished reference books, journals, scholarly papers, the internet, official reports, and documents was also used to gather secondary data. Direct observations were also done to collect data on the villages in general and to perceive the actual condition of the households and activities related to the forest [8].

Based on a one-month recall period, income data from wage, business, forest, and environmental sources were collected. On the other hand, data for the crop, livestock, fishing, and other income sources were based on three-month recall periods. Cash income from wages and business sources involved recording the amount earned. The recorded quantities of products (collected, bought, or sold) multiplied by the local prices were used to compute the cash and subsistence income from the forest, agriculture, environmental sources, fishing, and non-forest environmental sources. Based on the own-reported values of each household, cash equivalent values for subsistence goods and services were assigned. These values were then independently cross-checked by comparing them to the retail prices at main local and urban markets and comparing the average prices between sample villages. All incomes were approximated on an annual basis in order to obtain the total annual household income.

### 2.3. Income Calculations

The PEN technical guidelines were the basis for the income definition employed in this study [96]. The total income of a household is the sum of all cash income from different sources (such as crop and livestock production, gathering wild foods, and small-scale activities), as well as the monetary value of the output used by the household for subsistence (subsistence income was defined as the value of products consumed directly by the household or given away to friends/relatives) [13]. The total household income represented the net income produced by the sample of households under study. The total input value (such as fertilizer for agricultural production and veterinary supplies) was subtracted from the total output value during the accounting period. Because labor costs vary depending on the work, the cost of own labor was not taken into account in any income calculations. Multiple tasks being completed at once may cause an under- or overestimate of labor costs [13,97].

Seven main types of income were found: crop income, forest income, business income, livestock income, wage income, income from other sources, fish and non-forest environmental income. Below are descriptions of each of these categories:

Annual total household income:

$$Y = \sum_{i=1}^{n} (X_i),\qquad(1)$$

where Y denotes the annual total household income and $X_i$ is the income from source *i*.

We grouped the income sources into two main categories for the analysis: forest income and non-forest income (Crop Income, Business Income, Livestock Income, Wage

Income, Other Income, and Fish and non-forest environmental income). Our earlier research work [12] defined the specific methods for calculating income from each source based on the PEN technical standards [96].

*2.4. Data Analysis*

Both IBM® SPSS® Statistics 20 and Microsoft Excel® 2019 (IBM: Chicago, IL, USA; Microsoft Corporation: Washington, DC, USA) were used for data analysis. We proceeded by data entry, checking and correcting, computation of descriptive statistics, Kruskal–Wallis Test processing, cross-tabulate data for $\chi^2$ test of independence, and performing logistic regression analysis.

Variables were described using descriptive statistics, which included frequencies, proportions, central tendency measures, and dispersion measures. The Kruskal–Wallis Test is the non-parametric alternative to a one-way between-group analysis of variance [98]. This test assessed disparities in annual household income across households per income source. The $\chi^2$ test examined the relationship between income terciles and the level of forest dependence. Both Kruskal–Wallis Test, $\chi^2$ test and logistic regression analysis were performed using IBM® SPSS® Statistics 20.

2.4.1. Forest Dependence Index Calculation

The ratio of the total annual income from forests to the total annual income from all sources was used to determine how dependent a household was on forest resources [36,39,44,55,96].

$$RY_f = \frac{Y_f}{Y},\tag{2}$$

where $RY_f$ is the relative forest income, Y represents the total household income, and $Y_f$ is the total forest income.

For the purposes of this study, households with a positive income from forest-related activities are considered to be forest dependents. Because it is unclear what level of absolute income determines forest reliance, relative income is used instead. Absolute dependency is defined as the amount of forest products gathered, whereas relative dependence is the percentage of total income derived from forest products [2,99].

2.4.2. Determinants of Forest Dependence
Forest Dependence Model Specification

The logistic regression model was used in this study to analyze socio-economic and demographic factors' influence on forest dependence in the RTHD, DR Congo. The importance of a logistic model over an OLS (Ordinary Least Squares) model in addressing socioeconomic research, particularly dependence, has been recognized in several studies [2,17,39,44,62]. It is worthwhile to develop management plans that divide household status into "high forest dependency" and "low forest dependency" categories [39,55].

The forest dependence index was divided at the median in this study because there is no basis for categorizing it [100]. Another study also used this approach [2]. The median provides a better indicator of central tendency than the mean since high-value cases can significantly impact the mean but have little or no impact on the median [101]. Therefore, the median was taken as the cut-off threshold between the two levels of forest dependence (low and high). Other studies considered the average value as the cut-off point [44,55,68]. The forest dependence index's estimated median value for this investigation was 0.42. Therefore, households classified as having "low forest dependence" had a household forest dependence score below the cut-off of 0.42 and had a household forest income that made up less than 42% of their total income. On the other hand, those with a forest dependence index equal to or more than 0.42, whose income from forest accounted for equal to or more than 42% of their total income, were classified as having "high forest dependence." The variable forest dependency was measured as a dichotomous response occupying the value of 1 or 0, where 1 denotes high forest dependency and 0 means low forest dependency. Given

that the dependent variables are binary, it is suggested that the Binary Logistic Regression Model be used to identify the variables that affect households' reliance on forests [2,68,102].

The following was the specification of the model used to estimate the forest dependence:

$$ln\left(\frac{p_i}{1-p_i}\right) = \beta_0 + \beta_1 X_{1i} + \beta_2 X_{2i} + \ldots + \beta_k X_{ki}, \tag{3}$$

where $p$ is the probability of the outcome, $i$ denotes the $i$-th observation in the sample, $\beta_0$ is the intercept term, and an $\beta_1$, $\beta_2$, ..., $\beta_k$ are the coefficients related with each explanatory variable $X_1$, $X_2$ ... ... $X_k$ [39,44,55].

Variables Selection Rationale

The following explanatory factors were chosen and used to develop a theoretical association between forest dependency and household characteristics: the household head's gender, age, level of education, length of residency in the village, size of land holding, distance to the forest, and the non-forest income (the aggregate of incomes from crops, livestock, fish, and non-forest environmental, business, wage, and other sources) (Table 1). The variables have been selected primarily because they cut across social and economic domains, providing a comprehensive understanding of the pattern of household forest dependency [68]. Each explanatory variable is briefly described here, as well as the expected theoretical relationship with forest dependency.

**Table 1.** Description of the explanatory variables and their expected signs within the forest dependence model.

| Variables | Explanation | Expected Sign |
|---|---|---|
| Gender | The sex of the household head (0 if female, 1 if male). | Positive |
| Age | The age of the household head in years. | Negative |
| Education | Household head's level of education: we have assigned 0, 1, 2, and 3 to none (illiterate), primary, secondary, and university levels, respectively. | Negative |
| Household size | Family size of the household. | Positive |
| Length of residence | The number of years residing in the respective study area. | Positive |
| Landholding | Own land holding measured in hectare. | Negative |
| Distance to the forest | Distance to the forest measured in km. | Negative |
| Non-forest income | Aggregation of non-forest income sources in CDF (Congolese Franc: legal currency of the Democratic Republic of the Congo). | Negative |

Gender: Men and women gather forest products for various purposes [103]. While gathering firewood and medicinal plants is a shared activity, gathering thatching grass and wild fruits is a task reserved only for women. Cutting construction materials is a task that men only undertake. Men are more inclined than women to take the risk of entering the forest because it is illegal to extract forest products, and, in some situations, there may be a threat from wild animals. Therefore, households headed by men are more likely to rely on forest resources than households headed by women [2]. Then, a dummy variable was created for this variable, with 0 for females and 1 for males.

Age: Indicates how many years the head of the household has lived [68]. Any age group can be reliant on the forest. Younger people might rely more on forest resources than older ones. This is because young people may use the forests for various purposes, and that forest extraction activities are labor- and physically intensive [76,104]. Due to this reason, older people may reduce their dependence on forests [21,105]. Forest reliance is therefore considered to be inversely associated.

Education: shows the household head's greatest level of education. Education levels were divided into four categories: None (Illiterate), Primary, Secondary, and University.

Education increases prospects for various economic alternatives [55], moving people away from subsistence farming and other occupations [39,40,59,75,76,106,107]. Since educated people who work in public or private sectors can afford a modern lifestyle, their greater social status may also limit their participation in activities that rely on the forest. In this situation, household head education is expected to correlate negatively with forest dependency [2]. Education level was recorded as 0, 1, and 2 for None, Primary, Secondary and University levels, respectively.

Household size: reflects the total number of inhabitants living in a specific home depending on the forest [68]. Household size is typically strongly correlated with forest dependency because larger families may mobilize more of their families to engage in forest-based activities, leading to more resource extraction [7,39–41,47,76]. Furthermore, larger families have higher subsistence needs and more human resources to meet this demand [21,55,59,74]. Therefore, it is hypothesized that dependence on forests is directly related to larger households.

Length of residence: Indicates how long a household lived in the relevant study area [68]. Long-term residents are probably more familiar with the ecological composition, seasonal cycles, and structure of the forests, which leads them to gather more forest products [99,108]. Therefore, it is anticipated that the length of the residency will be directly related to dependence on the forest.

Landholding: Generally, households with more land should depend less on forests because they can access alternate agricultural resources for generating income [41]. Other studies [17,55] also discovered a statistically significant negative correlation between private land ownership and forest reliance. Therefore, it is assumed that there is an inverse association between landholding and reliance on forests.

Distance to the forest: The households that are relatively closer to the forest may rely more on its resources than other households [21,63]. As a result, the dependency of households on the forest is inversely related to distance from the forest.

Non-forest income: refers to the total household income from non-forest sources such as Crop, Livestock, Fish and Non-Forest environmental, Business, Wage and Other. Households with access to these income means are assumed to be less reliant on forest resources. The higher the non-forest income of households, the less dependent the household is [19,39,44,80,109,110]. According to certain studies [17,41,62,111,112], households with higher agricultural incomes rely less on forests because they choose to carry out agrarian agricultural activities in fields rather than forest extraction. Furthermore, previous research noted that people who engage in better off-farm activities rely less on forest resources due to their higher income from these sources [13,21,113]. According to a different study, dependence on the forest was inversely associated with the number of livestock units [55]. Hence, Non-Forest income is hypothesized to have an inverse relationship with forest dependence.

## 3. Results

### 3.1. Forest-Dependent Households Profile

A total of 127 households were surveyed, and 89.76% (*n* = 114) of those households were dependent on the forest and had positive forest income. Among them, 85.09% were headed by men, with just 14.91% by women. A total of 90.35% were married, 4.39% divorced, 2.63% widows/widowers, and 2.63% singles. The ages of household heads ranged from 21 to 79 years, with an average of 45.11 years. Regarding education, 1.75% had no formal education, while 45.61, 50.00, and 2.63% had completed their primary, secondary, and university education. The average household size was 8.99, varying between 1 and 25 family members. More than half (63.16%) of the forest dependents were not borne in their current residences. Most households (97.37%) lived in their own houses, while 2.63% were renting, with an average residency length of 26.44 years. The average land holding values and the forest distance was 1.57 ha and 2.83 km, respectively. In total, 90.35% of households reported owning endowment assets, with an average value of CDF 527,354.37 (cash savings, livestock, and domestic assets) (Table 2). Most livestock assets were sheep,

goats, chickens, ducks, and pigs. Furniture, bicycles, motorbikes, cell phones, solar panels, televisions, radios, pirogues, shotguns, CD and VCD players constituted the household assets. Concerning income sources, 36.45% of total annual income was from forest-based activities, while the remaining 63.55% from non-forest income sources (Crop income 32.03%, Livestock income 9.18%, Fish and Non-Forest environmental income 8.12%, Business 9.19%, Wage income 2.34% and Other income 2.70%) (Table 3).

**Table 2.** Forest-dependent households' characteristics.

| Variable | Items | Mean * | SD * | % |
|---|---|---|---|---|
| Gender | Male | | | 85.09 |
| | Female | | | 14.91 |
| Marital status | Married | | | 90.35 |
| | Divorced | | | 4.39 |
| | Widows/Widowers | | | 2.63 |
| | Singles | | | 2.63 |
| Age (years) | | 45.11 | 12.19 | |
| Education level | None | | | 1.75 |
| | Primary | | | 45.61 |
| | Secondary | | | 50.00 |
| | University | | | 2.63 |
| Household size | | 8.99 | 4.74 | |
| Place of birth | Current residence | | | 36.84 |
| | Another place | | | 63.16 |
| Length of residence | | 26.44 | 17.65 | |
| Residence | Owner | | | 97.37 |
| | Lodger | | | 2.63 |
| Landholding (ha) | | 1.57 | 1.02 | 100 |
| Walking distance to the forest (km) | | 2.83 | 1.83 | |
| Assets endowment (CDF *) | | 527,354.37 | 738,605.96 | 90.35 |
| Forest income (CDF *) | | 1,219,951.58 | 1,772,390.80 | 100 |
| Non-forest income (CDF *) | | 2,266,280.40 | 3,024,316.91 | 93.86 |

* CDF: Congolese Franc, which is the legal currency of the Democratic Republic of the Congo. The average annual exchange rate was USD 1 = 1851.00 CDF during the research period. * SD: Standard deviation. * Mean: Average deviation.

**Table 3.** Forest dependents involved in livelihood strategies, average annual income per income source (in CDF) and shares by income source.

| Income Components | Households Involved (%) | Average Income * | Standard Deviation * | Minimum | Maximum | Share (%) |
|---|---|---|---|---|---|---|
| Crop Income | 70.18 | 1,527,777.53 | 1,902,006.94 | 12,000.00 | 8,465,200.00 | 32.03 |
| Forest Related Income | 100 | 1,219,951.58 | 1,772,390.80 | 21,600.00 | 10,596,000.00 | 36.45 |
| Livestock Income | 38.60 | 796,204.55 | 1,306,169.86 | 48,000.00 | 8,220,000.00 | 9.18 |
| Fish and Non-forest env. Income | 45.61 | 595,592.31 | 1,225,917.60 | 4000.00 | 7,800,000.00 | 8.12 |
| Business Income | 13.6 | 2,337,600.00 | 3,428,072.28 | 180,000.00 | 10,200,000.00 | 9.19 |
| Wage Income | 7.02 | 1,114,000.01 | 991,997.69 | 80,000.04 | 2,280,000.00 | 2.34 |
| Other Income | 23.68 | 381,111.11 | 779,307.85 | 4000.00 | 4,000,000.00 | 2.70 |
| Total | | 3,347,074.41 | | | | 100.00 |

* The average and standard deviation for each activity were determined using data from the households involved, whereas the average total income was determined using data from all households.

### 3.2. Household Dependence on Forest Resources and Livelihoods Strategies

The average contribution of forest-related income to the total annual income was 36.45%. This contribution was greater than the respective incomes from crop, business, livestock, fish, non-environmental forests, other sources, and wage (Table 3). A significant difference in annual household income in terms of income components was found in the research area based on the Kruskal–Wallis test ($p < 0.5$). The main forest products that households harvested were Bushmeat, Firewood, Construction Materials, Fruits, Mushrooms, and Marantaceae Leaves (see our previous research article [12] for the detail regarding the contribution of these products to the Total Forest Income).

The level of reliance on forest resources was determined using the forest dependence index (FDI). It was found that household forest dependency indexes or ratios varied from 0.01 to 1, with a mean of 0.46 and a standard deviation of 0.30.

A Chi-square test for independence revealed a significant association between income terciles and the level of forest reliance status, $\chi^2$ (2, $n = 114$) = 7.66, $p = 0.02$. There is a significant difference in the household distribution of forest-dependent households by income terciles and level of forest dependency. In total, 68.42% of the low-income category, according to this distribution, were highly reliant on forest products. Comparatively, only 36.84% of the households in the high-income tercile were heavily reliant on forest goods, with the remaining households having a low reliance on forests (Table 4).

**Table 4.** Distribution of forest-dependent households by income tercile and level of forest dependence.

| Forest Dependence | Income Terciles | | | Total | N |
|---|---|---|---|---|---|
| | Low Income | Middle Income | High Income | | |
| Low | 31.58% | 50.00% | 63.16% | 48.25% | 55 |
| High | 68.42% | 50.00% | 36.84% | 51.75% | 59 |
| Total | 100.00% | 100% | 100% | 100% | 114 |
| N | 38 | 38 | 38 | | |

### 3.3. Determinants of Forest Dependence

We performed the Binary Logistic Regression Model to examine the predictive ability of the selected demographic and socioeconomic factors on household forest dependence. The model contained eight independent variables (gender, age of the household head, household size, education level of the household head, length of residence, landholding, distance to forest and non-forest income). We conducted preliminary assumption analyses to test for multicollinearity. All tolerance values for each explanatory variable were above the conventional cut-off point for tolerance (not less than 0.10), indicating that the multicollinearity assumption was not violated. This is also supported by the Variance Inflation Factor (VIF) values of each independent variable, which are well below the cut-off of 10 (not greater than 10).

The logistic regression model findings are presented in Table 5. The result of the likelihood ratio in the logistic regression test showed that the regression model containing all predictors was statistically significant, with Chi-Square statistics of 6.21. This result indicates that the explanatory variables in the model are significantly related to forest dependence. The explanatory variables were well selected and could be used to predict the dependent variable. The model explained between 44.2% (Cox and Snell R squared) and 58.9% (Nagelkerke R squared) of the variance on forest dependency and correctly classified 79.8% of the cases.

According to Table 5, Household size and Non-forest income demonstrated expected results when predicting forest dependence, returning *p*-values less than $\alpha = 0.01$ and less than $\alpha = 0.001$, respectively, suggesting that these two variables were statistically significant. The remaining explanatory variables (gender, age of the household head, education level of the household head, length of residence, landholding, and distance to forest) did not show significant results when explaining forest dependence.

**Table 5.** Determinants of household forest dependency (Logistic regression model).

| Variables | B | S.E. | Wald | df | Sig. | Exp(B) | 95% C.I. for EXP(B) | |
|---|---|---|---|---|---|---|---|---|
| | | | | | | | Lower | Upper |
| Gender | −0.808492 | 0.756246 | 1.142945 | 1 | 0.285031 | 0.445529 | 0.101195 | 1.961526 |
| Age | 0.009127 | 0.023753 | 0.147652 | 1 | 0.700789 | 1.009169 | 0.963264 | 1.057261 |
| Household size | 0.217239 | 0.076026 | 8.164828 | 1 | 0.004271 ** | 1.242641 | 1.070612 | 1.442313 |
| Education | | | 2.181337 | 3 | 0.535634 | | | |
| Education (primary) | 0.510423 | 3.820038 | 0.017854 | 1 | 0.893705 | 1.665996 | 0.000933 | 2973.692872 |
| Education (secondary) | 1.450625 | 1.415877 | 1.049686 | 1 | 0.305579 | 4.265781 | 0.265951 | 68.421897 |
| Education (university) | 0.689348 | 1.359257 | 0.257202 | 1 | 0.612049 | 1.992416 | 0.138796 | 28.601041 |
| Length of Residency | −0.013040 | 0.016275 | 0.641915 | 1 | 0.423018 | 0.987045 | 0.956056 | 1.019038 |
| Landholding | 0.259807 | 0.302570 | 0.737308 | 1 | 0.390525 | 1.296679 | 0.716613 | 2.346285 |
| Distance to the forest | 0.171960 | 0.151319 | 1.291419 | 1 | 0.255787 | 1.187630 | 0.882832 | 1.597659 |
| Non-forest income | −0.000002 | 0.000000 | 17.452248 | 1 | 0.000029 *** | 1.00000 | 0.999998 | 0.999999 |
| Constant | −1.522570 | 1.793741 | 0.720502 | 1 | 0.395979 | 0.218150 | | |

B: beta coefficients; S.E.: stand error; Exp(B): odd ratio (OR); **: $p < 0.01$; ***: $p < 0.001$; C.I.: Confidence Interval.

The results showed that the coefficient for household size was positive and the value of the odds ratio (OR = 1.243), suggesting that with the increase of 1 unit of household, the forest dependence will be around 1 time higher. It implies that large families depended more on forest resources to meet their subsistence requirements in the RTHD.

There was an inverse relationship between non-forest income and forest dependence, the odds of reporting high forest dependency decreased with the non-forest income (OR = 1.00). This suggests that an increase in non-forest income decreases forest dependency by a factor of 1.00, all other factors being equal. Forest reliance is reduced with agricultural, livestock, fish and non-environmental resources. Furthermore, people with better off-farm activities were less dependent on forest resources due to their higher income from their other sources.

On the other hand, the variables age of the household head, education level of the household head, landholding, and distance to the forest were not statistically significant but showed a positive association with forest dependency. In contrast, the gender of the household head and length of residence were not statistically significant but showed an inverse relationship with forest dependence.

The positive coefficient of the variable age of the household head signifies that the elderly households were likely to gather more forest products compared to youthful. Similarly, the positive coefficient of the household head's education level suggests that formally educated people depend more on forest resources. This is also for landholding and distance to forest variables, implying that households with more land tend to be dependent on the forest; an increase in distance to forest tends to increase forest dependency. Households were willing to go as far as needed to get the amount of these resources.

On the other hand, forest dependence and sex were negatively related. Therefore, the negative coefficient of sex implies that females were more dependent on forest resources than males. Similarly, the negative association between length of residence and forest dependence suggested that long-term residents were less reliant on forest resources than short-term residents.

## 4. Discussion

The study revealed that forest resources contributed up to 36.45% of the total annual income of forest-dependent households. This represents the proportion of cash and subsistence forest income. The proportions from Crop, Business, Livestock, Fish and Non-environmental forest, Other income and Wage incomes were 32.03%, 9.19%, 9.18%, 8.12%, 2.70%, and 2.34%, respectively. The household forest dependence indexes varied from 0.01 to 1 (Mean = 0.46, SD = 0.30). The distribution of forest-dependent households by Income Tercile and Level of Forest Dependency differs significantly (Table 4). The study

in Botswana found the mean of the forest dependency index to be 0.50, suggesting that households were moderately dependent on forest resources [68]. Research at Cameroon's Lobeke National Park revealed that 44.44% of total household income came from the forest, while 55.56% came from sources other than the forest [44]. The range of the dependence indices was 0.10 to 0.82. In addition, the level of dependency on forest products varied significantly among the wealth quintiles. In addition to agriculture, a different study found that collecting and selling forest products comprised 35 to 52.4% of household income in the South Province of Cameroon [44,114,115]. According to another study, this contribution to monthly household incomes for households engaged in NTFP extraction is over 40% [116]. These funds were often used to achieve different significant MDGs (Millennium Development Goals) [117]. While revealing different proportions, these results are consistent with our study's findings in the RTHD.

The study conducted in India revealed that the forest reliance ratios ranged from 0.3 to 0.5. The income derived from the forests contributed 43% to the total income of all sampled households [55]. A similar study showed that the contribution of forests to the total household income was 39%, and the mean forest resource dependency was 0.47 [17]. In Northern Pakistan's Chaprote Valley, forest income contributed 32% of the total annual household income, while the remaining 68% was from non-forest income. The computed average forest dependency had a value of 0.55 [21]. Forest income was the second most significant source of income in Ethiopia's eastern highlands, contributing an average of 32.6%. The percentages of the total household income related to crop production, livestock, off-and non-farm activities, and woodlots were 40.7%, 13.6%, 11.4%, and 1.7%, respectively [73]. These results are in line with investigations carried out in other forested regions [3,118,119]. According to a survey from Zimbabwe, which is comparable to the current study, forest income made up around 35% of total annual household income [13]. Additionally, an assessment revealed that the forest's primary source of revenue for the rural poor was wood fuel [36]. Mean forest income per household varied with wealth status. Poor households earned significantly more income. Poor households' reliance on forests was noted in several studies [6,46].

All these findings are in line with those revealed in our study, suggesting that households were somewhat more dependent on the natural forest resources than the other income sources.

With regard to the determinants of forest dependence, our findings showed that the variables household size and non-forest income significantly influenced the level of forest dependency in the RTHD. The coefficient for household size was positive, suggesting that the increase of 1 unit of the household size increases forest dependence. This suggests that because the extraction of forest resources requires a lot of labor, large families are more dependent on them. Large households need more firewood and other forest resources for their daily needs. Other investigations observed similar results in other forest areas [3,7,17,47,61,76,106]. In contrast, other studies reported contradictory results [44,68]. This unexpected result may be explained by variations in demographic profiles [68]. The positive association between household size and reliance on forests has also been documented elsewhere [2,40,41,62,105,120].

Non-forest income was negatively associated with forest dependency, suggesting that households having income from non-forest activities tended to reduce their dependence on forest resources in the study area. Similar studies elsewhere also reported that the higher the non-forest income of the household, the less dependent the household is [19,39,44,71,80,110]. Other findings reported a negative association between employment and forest dependency [3,40,44,68]. Employed households were less dependent on forest products. Several other studies revealed this negative association. The livestock income was statistically significant and negatively associated with forest dependency elsewhere [17,55,59,61]. Similarly, another study reported that agriculture had an inverse relationship with forest dependency [17]. Due to the large contribution of agricultural products to family economies in the Tharawady district of Myanmar, the findings showed that the

income from agriculture had a strongly negative association with forest dependency [112]. This suggests that household heads with higher agricultural income levels relied less on forest goods [21,41,62,111]. Low-income households with little agricultural income might rely increasingly on the forest [13,41,107]. Other research documented the adverse effects of off-farm income on forest dependency. The households that derived a larger portion of their income from non-farm sources (businessman income, government servant income, and daily wage labor income) relied less on forest resources [3,13,17,21,29,75,113]. Furthermore, such activities increased productivity and income, which could divert households from forest extraction [17]. In contrast, several studies discovered a positive association between cattle population and reliance on forests. An increase in livestock tends to enhance the likelihood of becoming a household heavily dependent on the forest [21,55].

Age of the household head, level of education of the household head, landholding, and distance to the forest were not statistically significant but had a positive relationship with forest dependency in the RTHD. The positive coefficient of the variable age of the household head signifies that the elderly households were likely to gather more forest products compared to youthful. Similar findings were reported in another study [17]. This was contrary to findings reported by other studies, which showed an inverse relationship between the age of household head and forest dependence, implying that the youth were more likely to rely on forest products than the older people [44,62,68,76,104]. Furthermore, younger households are trapped in poverty due to limited alternative economic opportunities [121]. The household head's education level was positively associated with forest dependence, suggesting that formally educated people depended more on forest resources. Similar findings have been reported in other studies [17,62]. Although this result is somewhat counterintuitive, this observation may be due to a lack of variation in education among households. Additionally, given the lack of off-farm employment alternatives in rural areas, educated individuals with a higher knowledge of forest products may have a competitive advantage over others who lack such knowledge [62]. Other investigations revealed a negative association between education level and dependence on forests. Household dependency on forest products decreases as education levels rise [68]. This may be explained by the assumption that people with formal education have various livelihood options that may provide higher returns than those related to the forest [39]. This result is in line with research from elsewhere [2,7,20,21,41,73,76,97,106,122].

Landholding was positively associated with the forest dependence level, suggesting that households with more land tended to be more dependent on the forest. It seems to be counterintuitive. The possible reason would be that forest products are crucial in the context of the rural households in the RTHD, despite the increase in landholding size. The other reason could be that landholding size had no significant disparities between forest-dependent households. Another study reported this positive association [44]. Similar studies elsewhere reported contrary findings showing the negative association between landholding and level of dependence on forest, implying that households with more substantial landholdings were less dependent on forest resources [8,17,55,62,123,124]. Besides, the distance to the forest positively affected the dependence on the forest, implying that an increase in distance tended to increase the forest dependence. Fuel wood and other important forest products constitute basic needs for forest dependents and cannot be easily substituted by others in the RTHD. Another study reported this positive and significant relationship [59]. Households will go as far as needed to get the fuel wood. Other studies showed an inverse relationship, indicating that households far from the forest and with more landholding are less inclined to forest resources extraction [3,5,17,61,63,73,106,125].

Finally, although the household head's gender and length of residency were not statistically significant, they did show an inverse association with forest dependence. According to the research, females were more reliant on forest resources than males in the RTHD. Findings from several investigations [38,44,68,126] have supported this. Being female has a positive and significant impact on the amount of forest income in Ethiopia, according to various research [73,123]. Compared to men, women are more dependent

on the forest because they lack the resources and other assets needed to engage in intense agricultural production or non-farm activities to boost their income [73]. Studies conducted elsewhere, however, revealed that men were more likely than women to rely more on forests and other natural resources [39,62]. In the case of the length of residence, our findings showed that long-term residents were less reliant on forest resources than short-term residents. This was in contrast to other studies' findings reported elsewhere [68,99].

## 5. Conclusions

The study aimed to identify the main determinants driving the dependence of rural households on forest resources in the Rubi-Tele Hunting Domain, DR Congo. We, therefore, determined the forest dependence level and identified the demographic and socioeconomic factors influencing the forest dependence level. Data were gathered through structured household-level surveys, focus group discussions, and key informant interviews. The results revealed that forest use was important for rural households (forest dependents), and the related pattern was driven by the forest-dependent households' demographic and socioeconomic characteristics. The cash and subsistence forest income proportion represented 32.46% of the total annual forest-dependents income. The forest dependence index varied from varied from 0.01 to 1 (Mean = 0.46, SD = 0.30). The distribution of forest-dependent households by Income Tercile and Level of Forest Dependence differs significantly. This distribution showed that 68.42% of the low-income group highly depended on forest products. In comparison, only 36.84% of the forest-depend households in the high-income tercile were highly dependent on forest products. The results from the binary logistic regression revealed that household size and non-forest income had a statistically significant impact on the level of forest dependence. Large families tended to depend more on forest resources. On the other hand, the non-forest income was inversely related to forest dependence, implying that forest-dependent households with non-forest income sources were less dependent on forest resources extraction in this protected area. The non-forest income included incomes from Agriculture, Livestock, Fish and Non-Forest environment, and Off-Farm (Business, Wage and Other income).

In light of the findings revealed in this study, forest resources are an important component of the households' activities in forest zones, particularly in protected areas. Therefore, the factors driving the pattern of forest use need to be considered in designing the management of forest resources, which ensures the balance between the needs of forest dependents and conservation in the RTHD, DR Congo. In the face of rapidly growing human populations in and around the bio-diverse regions of tropical forests, sustainable use of timber and non-timber forest products is a great challenge. In this perspective, a participative management model could be encouraged. Furthermore, establishing favorable policy incentives could help reduce the households' dependence on forest resources in the RTHD, DR Congo. Controlling household size while providing incentives in terms of alternative means for income generation is highly recommended. The development of the tourism sector is also highly recommended. This could help diversify alternative livelihood opportunities, which can sustain the local socio-economic development in the RTHD. Other energy sources are crucial to decrease the heavy reliance on the consumption of firewood. Communities should be educated on the importance of the RTHD through environmental education.

**Author Contributions:** Conceptualization, R.K.M. and G.T.; methodology, R.K.M. and G.T.; software, R.K.M.; validation, R.K.M. and G.T.; formal analysis, R.K.M.; investigation, R.K.M.; resources, R.K.M., G.T. and P.M.M.; data curation, R.K.M.; writing—original draft preparation, R.K.M.; writing—review and editing, R.K.M., G.T. and P.M.M.; visualization, R.K.M., G.T. and P.M.M.; supervision, G.T.; project administration, R.K.M. and G.T.; funding acquisition, G.T. All authors have read and agreed to the published version of the manuscript.

**Funding:** This research was funded by the National Social Science Foundation Project of China, grant number: 21BGJ066.

**Data Availability Statement:** Not applicable.

**Acknowledgments:** The authors express their gratitude to everyone who helped make this research a success.

**Conflicts of Interest:** The authors declare no conflict of interest.

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
