# Peer review of "Identifying Socioeconomic Determinants of Households’ Forest Dependence in the Rubi-Tele Hunting Domain, DR Congo: A Logistic Regression Analysis"

_forests, doi:10.3390/f13101706_

Round 1
Reviewer 1 Report
(1)Line 195-196, how to select the 127 samples? please add it.
(2)Line 300-306, how to define the Variables, for example, gender, age, why select these variables? Or What is the basis for selecting variables ? please add the explanation.
(3)Line 445 OR=0.99 should OR=1.00
Author Response
"Please see the attachment."

Reviewer 2 Report
In this manuscript the authors state the Identifying socioeconomic determinants of households' forest dependency in the Rubi-Tele Hunting Domain, DR Congo: A Logistic Regression Analysis.
Reviewing this manuscript made me think about my role as a reviewer and what it is that can potentially make a manuscript a value contribution to the literatures within its field. Though this paper described some interesting issues about Identifying socioeconomic determinants of households' forest dependency in the Rubi-Tele Hunting Domain, they are not supported by empirical evidence well and are too weak to pass the review.
In the introduction, should the logical sequence between paragraphs be reorganized to make the reading of this part smoother?
The article used data from 127 samples. Although the data sources were described in detail, the sample size was small and whether it was representative.
The article divides the income into three levels, then what is the basis and standard for such a division.
Is there an endogenous problem between Non-forest income and forest dependence? The higher the forest dependence, the lower the Non-forest income..
Explanations should be given for the variable results that are contrary to the expectations in the introduction, literature review and variable prediction (why are these variables contrary to other research results and their own predictions? ).
In the result, the robustness test is lacking. In the regression model, education is analyzed as a continuous variable, so why are other variables such as age not analyzed as a continuous variable?
The research content is not innovative, just like the analysis results, only two variables are significant. So whether to consider adding other core variables?
Author Response
"Please see the attachment."

Round 2
Reviewer 2 Report
This study is interesting for me, for I note that there are rarely literatures concerning forest dependent communities in African nation.
Structural thinking, the Ms. is well organized with contents of introduction, methods, results, discussion, and conclusion.
I have two specific comments for authors as references.
1) Since this study is conducted in the region of a hunting domain, the issues related to income of hunting and relevant activities had better to be taken into account. Otherwise, the Ms. could not reveal a complete picture of the local peoples' income.
2) The uniqueness of this study, viz., the knowledge gap raised by the authors, is not convincing enough to me. The authors usually will not insist that a study in a new region is unique. The uniqueness should be well reflected by theoretic innovation.
Author Response
"Please see the attachment."
